# Insights into Early Recovery from Influenza Pneumonia by Spatial and Temporal Quantification of Putative Lung Regenerating Cells and by Lung Proteomics

**DOI:** 10.3390/cells8090975

**Published:** 2019-08-26

**Authors:** Joe Wee Jian Ong, Kai Sen Tan, Siok Ghee Ler, Jayantha Gunaratne, Hyungwon Choi, Ju Ee Seet, Vincent Tak-Kwong Chow

**Affiliations:** 1Department of Microbiology and Immunology, Yong Loo Lin School of Medicine, National University of Singapore, Singapore 117545, Singapore; 2Department of Otolaryngology, National University of Singapore, Singapore 119228, Singapore; 3Institute of Molecular and Cell Biology, Singapore 138673, Singapore; 4Department of Medicine, National University of Singapore, Singapore 117599, Singapore; 5Department of Pathology, National University of Singapore, Singapore 119074, Singapore

**Keywords:** influenza, pneumonia, lung regeneration, stem cells, comparative quantification, P63, KRT5, proliferating alveolar type II cells, proteomics

## Abstract

During influenza pneumonia, the alveolar epithelial cells of the lungs are targeted by the influenza virus. The distal airway stem cells (DASCs) and proliferating alveolar type II (AT2) cells are reported to be putative lung repair cells. However, their relative spatial and temporal distribution is still unknown during influenza-induced acute lung injury. Here, we investigated the distribution of these cells, and concurrently performed global proteomic analysis of the infected lungs to elucidate and link the cellular and molecular events during influenza pneumonia recovery. BALB/c mice were infected with a sub-lethal dose of influenza H1N1 virus. From 5 to 25 days post-infection (dpi), mouse lungs were subjected to histopathologic and immunofluorescence analysis to probe for global distribution of lung repair cells (using P63 and KRT5 markers for DASCs; SPC and PCNA markers for AT2 cells). At 7 and 15 dpi, infected mouse lungs were also subjected to protein mass spectrometry for relative protein quantification. DASCs appeared only in the damaged area of the lung from 7 dpi onwards, reaching a peak at 21 dpi, and persisted until 25 dpi. However, no differentiation of DASCs to AT2 cells was observed by 25 dpi. In contrast, AT2 cells began proliferating from 7 dpi to replenish their population, especially within the boundary area between damaged and undamaged areas of the infected lungs. Mass spectrometry and gene ontology analysis revealed prominent innate immune responses at 7 dpi, which shifted towards adaptive immune responses by 15 dpi. Hence, proliferating AT2 cells but not DASCs contribute to AT2 cell regeneration following transition from innate to adaptive immune responses during the early phase of recovery from influenza pneumonia up to 25 dpi.

## 1. Introduction

The lung is a vital organ responsible for the gaseous exchange between the body and external environment. Exposure to airborne debris, pathogens, and chemicals may lead to acute or chronic injury of the lung tissues [1]. Influenza virus infection is one of the most common global infectious diseases, afflicting millions of people worldwide during influenza epidemics [2]. During seasonal influenza infection, the epithelial and immune cells in the upper respiratory tract are targeted [3,4], resulting in inflammation and death of these cells. In healthy individuals, the infection is typically self-limiting, and they usually recover within 7 days [5]. However, pandemic strains of influenza virus (such as the 1918-H1N1 Spanish Flu and the 1957-H2N2 Asian Flu), and occasionally seasonal strains, can infect the lower respiratory tract, resulting in the inflammation of alveolar air sacs, giving rise to influenza pneumonitis. This condition may cause diffuse alveolar damage, necrosis and hemorrhage of the alveolar epithelium [6,7], leading to acute respiratory distress syndrome (ARDS) and even death. While the damage to the lungs may be severe, patients who survive are usually able to achieve partial or full recovery within two years [8,9,10], demonstrating the ability of the lungs to regenerate following pneumonia.

Stem cells in the lower respiratory tract have been described to participate in lung repair and regeneration. These include the basal cells in the trachea and bronchus with dual expression of Trp63 (P63) and cytokeratin 5 (KRT5) [11], and the bronchio-alveolar stem cells (BASCs) in the bronchioalveolar duct junction (BADJ) that express uteroglobin (CC10) and surfactant protein C (SPC) [12]. In the alveoli, two populations of putative stem cells have been identified. The first population of alveolar stem cells is thought to be proliferating alveolar type II (AT2) cells that express SPC [13,14,15,16]. Recently, it was suggested that these proliferating AT2 cells are the main stem cells in the lungs for replacing the alveolar epithelium during influenza pneumonia [16]. However, another distinct group of stem cells has been documented in the regenerating lung following sub-lethal infection of a mouse-adapted H1N1 influenza virus strain (PR8). Similar to the basal cells in the trachea, these cells known as distal airway stem cells (DASCs) [17] or lineage-negative epithelial progenitors (LNEPs) [18], also express P63 and KRT5. DASCs are believed to be of a different lineage from the tracheal basal cells, and are shown to differentiate into alveolar epithelial cells in a mouse lineage-tracing model [17,19]. However, other studies suggested that these DASCs did not conclusively give rise to the regeneration of the alveolar epithelial cells destroyed following ARDS [18,20].

Studies that concurrently compare both populations of putative stem cells in the lungs during early recovery from influenza pneumonia are lacking, and we sought to analyze the relative contribution of these two cell types to early alveolar regeneration of the distal airways. Thus, to shed light on such early events, we employed a wild-type BALB/c mouse model of pneumonia induced by PR8 infection to simultaneously delineate and quantify the short-term spatial and temporal changes of DASCs and proliferating AT2 cells in the lungs from the time of infection till early recovery from influenza pneumonia. We found that by 25 days following infection, when mice regained their physical healthy state, their lungs still exhibited signs of damage but the DASCs had not differentiated into alveolar epithelial cells. On the other hand, we observed that AT2 cells proliferated to replace the AT2 cells that were destroyed during infection, suggesting that proliferating AT2 cells contribute positively to regeneration of alveolar epithelium [16]. In addition, proteomic analyses of global lung proteins revealed prominent innate immune responses at 7 dpi, which shifted towards adaptive immune responses by 15 dpi. In summary, our findings provide a better understanding of the early events of the cellular and molecular regenerative processes in the lungs in response to injury from influenza viral infection. 

## 2. Materials and Methods

### 2.1. Influenza Virus and Infection of Animals

Animal experiments were carried out with 6–8-week-old BALB/c mice in an animal BSL2 facility, under a protocol (Protocol number: R15-0014) approved by the Institutional Animal Care and Use Committee, National University of Singapore on 4^th^ March 2015. Mice in each group were infected on the same day with influenza A/H1N1/Puerto Rico/8/1934 (PR8) at a sub-lethal dose of 20 plaque-forming units (PFU) in 25 μL of PBS via intra-tracheal administration. Infection was carried out using the same diluted virus stock by one operator to ensure that each mouse received a similar virus inoculum. Control mice were mock-infected with sterile PBS, and euthanized 7 days later. The animals were anesthetized by intra-peritoneal injection of ketamine (75 mg/kg body weight) and medetomidine (1 mg/kg), and anesthesia was reversed by intra-peritoneal atipamezole (1 mg/kg). From 5 until 25 days post-infection (dpi), three mice were euthanized every two days and their lungs harvested. To determine if the mice were successfully infected, RNA was harvested from part of the lungs of representative mice at 5 dpi using the miRNeasy kit (Qiagen, Hilden, Germany). Extracted RNA (50 ng per sample) was converted to cDNA using MMLV reverse transcriptase (Promega, Madison, WI, USA). Quantitative real-time PCR (qPCR) was then performed using influenza virus NS1 gene-specific primers (forward 5′-CCATCAATTACCTGCCCCTA-3′; reverse 5′-GGAAGAGAAGGCAATGGTGA-3′) for 50 cycles each at 95 °C for 10 sec, 50 °C for 5 sec, and 72 °C for 5 sec. A standard curve was derived using RNAs of known viral loads (R^2^ = 0.9869). Viral NS1 RNA was detectable in the lungs of all three mice at 5 dpi (data not shown).

### 2.2. Tissue Processing and Histologic Staining

Harvested lungs were kept overnight in 4% paraformaldehyde, before being paraffinized in an ATP-700 automatic tissue processor (Histo-Line Laboratories, Milan, Italy). Lungs were then embedded in paraffin blocks, and cut with a RM2255 microtome (Leica Microsystems, Wetzlar, Germany) at 4 μm thickness. Consecutive sections per block were selected and then placed on polysine^TM^ slides (Thermo Fisher Scientific, Waltham, MA, USA) for subsequent immunofluorescence or hematoxylin and eosin (H&E) staining. For the latter, slides were placed twice in clearene, before being rehydrated in decreasing ethanol concentration. The slides were then immersed in concentrated hematoxylin for 10 min, before being counterstained with 1% eosin and mounted with permount medium.

### 2.3. Immunofluorescence Staining

The paraffin slides were rehyrated under a decreasing ethanol concentration, heated in 10 mM sodium citrate buffer at 95 °C for 30 min for antigen retrieval, and cooled to room temperature. The slides were then permeabilized with 0.5% Triton-X and blocked with blocking buffer (5% donkey serum, 1% bovine serum albumin (BSA), 0.1% Triton-X, 0.05% Tween-20 in Tris-buffered saline) for 2 h at room temperature. Primary antibodies used for immunofluorescence included: P63 (1:600, ab124762, Abcam, Cambridge, UK); KRT5 (1:50, LS-C22715, LifeSpan BioSciences, Seattle, WA, USA); PCNA (1:50, sc-9857, Santa Cruz Biotechnology, Texas, USA); SPC (1:50, sc-13979, Santa Cruz Biotechnology, Dallas, TX, USA); PDPN (1:50, sc-23564, Santa Cruz Biotechnology); and CC10 (1:50, sc-9972, Santa Cruz Biotechnology). Primary antibodies were diluted with 1% BSA, and incubated overnight at 4 °C. The slides were washed with Tris-buffered saline with 0.05% Triton-X (TBST) before incubating with either TBST-diluted donkey anti-goat alexa-fluor 488 (1:200, A11055, Thermo Fisher Scientific), donkey anti-rabbit alexa-fluor 555 (1:200, A31572, Thermo Fisher Scientific) or goat anti-guinea pig alexa-fluor 488 (1:200, ab150185, Abcam) at room temperature for 1 h. The slides were then washed, incubated in 0.1% Sudan black in 70% ethanol for 20 min to reduce autofluorescence [21], before further washing and incubating with 15 mM DAPI (in TBS) for 5 min. The slides were finally washed before mounting with ProLong Diamond Antifade mountant (Thermo Fisher Scientific, Waltham, MA, USA), and kept at 4 °C until imaging.

### 2.4. Imaging and Quantification of H&E and Immunofluorescence Slides

H&E and immunofluorescence slides (entire lung tissue) were scanned at 20× magnification with the TissueFAXS PLUS system (TissueGnostic, Vienna, Austria). The H&E and immunofluorescence images of the same lungs were overlaid together using PhotoShop CC 2019 image editor (Adobe, San Jose, CA, USA). The damaged area (DA) was demarcated as represented in Appendix A, and the KRT5-positive area as a percentage of the DA was quantified. The areas were measured using the open-source Fiji app (based on ImageJ; University of Wisconsin, Madison, WI, USA) and calculated as a ratio. Quantification of KRT5 intensity: FITC channel of the areas expressing P63, KRT5 in the immunofluorescence images were demarcated and measured using Fiji intensity measurement tool. Quantification of the proportion of DASCs expressing both P63 and KRT5 markers: Total cells with dual expression of P63 and KRT5, and total KRT5-positive cells were manually counted with the Fiji cell counter tool and expressed as a ratio. Quantification of cells expressing SPC and/or PCNA: Damaged areas (DA), boundary areas (BA) and undamaged areas (UA) were demarcated as exemplified in Appendix A. Five random fields (at 20× magnification) in DA, BA and UA were then selected from each lung, and total cells with dual expression of SPC and PCNA, or cells positive for SPC only were manually counted using Fiji cell counter tool.

### 2.5. Global Lung Protein Expression by Proteomic Analysis

BALB/c mice were infected as described above. Two mice each were euthanized at 7 and 15 dpi, and their lungs homogenized in urea lysis buffer. Uninfected mouse lungs served as the control. The homogenized lungs from each time-point were pooled, and subjected to in-solution digestion. Briefly, proteins underwent reduction (5 mM dithiothreitol) and alkylation (10 mM indole-3-acetic acid). The proteins were then subjected to overnight digestion with LysC (Wako, Osaka, Japan) at a protease-to-protein ratio of 1:100 at 37 °C, followed by sequencing grade trypsin (Promega) at a protease to protein ratio of 1:50 at 37 °C for 4 h. The resultant peptide solutions were acidified with 1% trifluoroacetic acid and desalted using 3M Empore 1-mL C18 SPE cartridge (Sigma-Aldrich, St. Louis, MO, USA). Desalted peptides were eluted with 0.1% formic acid in 80% acetonitrile/water. Tandem Mass Tag™ 6-plex (TMTsixplex™, Thermo Fisher Scientific, Waltham, MA, USA) reagents were used to label the desalted peptides. The TMT-labeled peptides were further fractionated using Agilent 3100 OFFGEL Fractionator (Agilent, Santa Clara, CA, USA) on a 13-cm IPG strip of pH 3–10. The 12 fractionated samples were desalted using self-packed C18 stage tips and analyzed on an EASY-nLC 1000 (Thermo Fisher Scientific) coupled to an Orbitrap Fusion mass spectrometer (Thermo Fisher Scientific). The peptides were resolved and separated on a 50-cm analytical EASY-Spray column equipped with pre-column over a 180-min gradient ranging from 8 to 40% of 0.1% formic acid in 95% acetonitrile/water. Survey full-scan MS spectra (*m*/*z* 310–1800) were acquired with a resolution of 120k, an AGC target of 2 × 10^5^ and a maximum injection time of 50 ms. MS2 scans were acquired with quadrupole isolation mode with CID activation using ion trap detector of an AGC target of 3 × 10^4^, a maximum injection time of 35 ms, and filtered with TMT isobaric tag loss exclusion. MS3 scans selected synchronous precursor using HCD activation of 65% collision energy and resolution of 60,000 for mass scan range of 100–500 with AGC target of 1 × 10^5^ and maximum injection of 120 ms. Raw mass spectrometry data were analyzed using the MaxQuant software [22]. Differential protein expression analysis was performed using the mapDIA tool [23], and functional enrichment analysis was achieved by an in-house implementation of hypergeometric test-based pathway enrichment tool and a combination of Gene Ontology [24] and Consensus Pathway DB [25]. Protein fold-change was calculated as the ratio of protein abundance at 7 or 15 dpi, with reference to uninfected mice. Each protein was considered differentially abundant if the reported false detection rate (FDR) was lower than 0.01, and each protein was quantified by at least 5 peptides. 

## 3. Results

### 3.1. Spatial and Temporal Distribution of DASCs Following Infection to Early Recovery

DASCs were not observed in uninfected lungs of control mice and at 5 dpi (Figure 1A,B). These cells began to be observed at 7 dpi, initially restricted only to the bronchioles before budding out from these at 9 dpi as small pods expressing KRT5 but not P63 (Figure 1C,D). By 11 dpi, DASCs expressing both P63 and KRT5 could be seen as distinct pods radiating outwards from the bronchioles (Figure 1E), with lumen formation commencing at 13 dpi (Figure 1F). Increasingly widespread lumen formation was observed in the DASC pods from 15 to 17 dpi (Figure 1G,H), before beginning to flatten out and to line the alveolar spaces at 19 dpi (Figure 1I). From 21 dpi onwards, DASCs no longer displayed a pod-like structure (Figure 1J–L) and the average intensity of KRT5 expression weakened by 30% from 21 to 25 dpi (Appendix A). The distribution of the DASCs over time is summarized in Table 1. The appearance of the DASCs in the lungs at 9 dpi coincided with the greatest weight loss of infected mice, after which their weight began to recover (Appendix A), implying that DASCs were associated with the recovery of the mice. A previous study showed that DASCs were found only in the damaged region of the lungs following influenza pneumonia [16]. Hence, we segregated the undamaged area (UA) and damaged area (DA) of the lungs (Appendix A). Indeed, we also observed DASCs only in the DA (Appendix A), and these DASCs were maximal at 21% of the total DA at 21 dpi (Appendix A). Moreover, we also noticed that not all DASCs co-express P63 and KRT5, i.e., only 3–10% of total KRT5-positive cells co-expressed P63 (Appendix A).

### 3.2. DASCs Do Not Conclusively Differentiate to Alveolar Epithelial Type II cells by 25 dpi

DASCs are reported to be adult stem cells in the lungs, and shown to differentiate into alveolar epithelial cells in vivo and ex vivo [17,19] using the KRT5 and KRT14 mouse lineage tracing model. Hence, we sought to analyze the expression of alveolar markers (type II cells by SPC staining, type I cells by podoplanin or PDPN staining) in the DASCs of the lungs over time. Stem cells are defined as a group of cells that are able to self-renew, proliferate, and differentiate into other cells. Indeed, the DASCs in the lungs were proliferating as evidenced by their high expression of proliferating cell nuclear antigen (PCNA) at 13 dpi, although this expression decreased significantly by 25 dpi (Figure 2A–F). Interestingly, we did not observe expression of SPC in the DASCs (Figure 2D–G), suggesting that DASCs were not differentiating to AT2 cells by 25 dpi. To ensure the integrity of our immunofluorescence assay, we stained an undamaged area of the lung which showed positive SPC immunostaining as expected (Figure 2H). However, we observed patchy expression of alveolar type I cell marker (PDPN) in the lumens of DASCs from 19 dpi, although it was detected in only 1% of DASCs by 25 dpi (Appendix A). In addition, it was also noted that the DASCs did not express the club cell marker CC10 (Appendix A). As expected, in mock-infected control mice, PDPN was widely expressed in uninfected lungs, while CC10 was detected in the bronchioles of uninfected lungs (Appendix A). Taken together, our data suggested that DASCs did not conclusively differentiate into alveolar epithelial cells by 25 dpi, in agreement with a previous study [20].

### 3.3. Proliferating Alveolar Type II Cells Increase During Recovery From Influenza Pneumonia

Proliferating AT2 cells constitute another population of cells described to give rise to AT2 cells following various injury models such as influenza, bleomycin and oxidants [13,16,26,27,28]. These cells displayed enlarged nuclei similar to DASCs, but did not organize into pod-like structures beside bronchioles. Instead, these proliferating AT2 cells existed either as a small bundle of cells, or as individual cells within infected lungs, but only a relatively low number of these cells were present in control uninfected lungs (Appendix A). We quantified the total number of proliferating AT2 cells (co-expressing both SPC and PCNA) in three separate areas of the lungs, i.e., undamaged area (UA), damaged area (DA) and boundary area (BA) (Appendix A). We observed that proliferating AT2 cells in BA began to increase from 7 dpi, peaked at 13–17 dpi, before decreasing until 21 dpi (Figure 3A,B). In the UA, these cells increased from 11 dpi, reached a peak at 13–15 dpi, before decreasing until 19 dpi (Figure 3D,E). In the DA, the number of proliferating AT2 cells increased from 11 dpi, albeit at a slower rate compared to BA and UA, peaking only at 19 dpi—although these cells persisted at relatively higher numbers even by 25 dpi (Figure 3G,H). Interestingly, the trend of the proliferating AT2 cells in the DA appeared to mirror the temporal appearance of KRT5-positive DASCs in the DA (Appendix A). When we analyzed expression of SPC only, there was a marked elevation of SPC expression in the BA at 19 and 25 dpi (Figure 3B), and in the DA at 25 dpi (Figure 3H), compared to 5 dpi when there was little to no SPC expression. Indeed, there were significantly more SPC cells in all three areas by 25 dpi (Figure 3C,F,I). Taken together, the decrease in proliferating AT2 cells expressing both PCNA and SPC from 19 dpi onwards, and the increase in only SPC-expressing cells at 25 dpi suggested that the AT2 cells proliferated following viral pneumonia injury to repopulate themselves, eventually giving rise to the new AT2 cells from 19 dpi. While both proliferating AT2 cells (Figure 3G) and DASCs (Appendix A) were found in DA, they were distinctly separate and did not overlap in terms of their spatial distribution (Figure 2G).

### 3.4. Global Protein Expression of Mouse Lungs at 7 and 15 dpi

Given that we did not observe any lung recovery at 7 dpi (Table 1), and that the DASC pods appeared to give rise to lumens (Table 1, Figure 1G) and that proliferating AT2 cells peaked at 15 dpi (Figure 3A,D), we proceeded to analyze the global protein expression of the lungs at 7 and 15 dpi to understand the molecular processes that operate at these two time-points. The mass spectrometry proteomics data have been deposited in the ProteomeXchange Consortium (http://proteomecentral.proteomexchange.org) via the PRIDE partner repository [29] with the dataset identifier PXD014967 and 10.6019/PXD014967. Compared to uninfected lungs, the top 5 up-regulated proteins at 7 dpi were IFI44 (29-fold), IIGP1 (9.3-fold), IFIT1 (7.2-fold), APCS (7.2-fold) and HP (7.1-fold) (Figure 4A). IFI44, IIGP1 and IFIT1 are interferon-induced proteins, whereas APCS and HP are acute phase proteins (Table 2). From the Gene Ontology (GO) analysis of the entire set of up-regulated and down-regulated proteins, we noted that processes related to innate immunity were among the top hits at 7 dpi (Appendix A). The top 5 down-regulated proteins at 7 dpi were CBR2 (3.4-fold), COL1A1 (3.1-fold), SFTPA1 (2.9-fold), LAMC2 (2.9-fold) and SFTPB (2.8-fold) (Figure 4A). CBR2, SFTPA1 and SFTPB are markers of AT2 cells, while COL1A1 and LAMC2 are subunits of type I collagen and laminin, which are components of the lung extracellular matrix (Table 2). Their down-regulation is congruent with the loss of alveolar epithelial cells that was already evident by 5 dpi (Figure 3A,D,G), and damage to the extracellular matrix (Appendix A), both of which may be attributed to infection and the up-regulation of innate immune processes.

At 15 dpi, the top 5 up-regulated proteins were IGH-1A (28-fold), IGH-3 (9.5-fold), IFI44 (9.0-fold), TNC (9.0-fold), and TAPBP (6.5-fold) (Figure 4B). The transition from acute inflammation reflected in 7 dpi to the adaptive immune response was indicated by the up-regulation of immunoglobulin gamma, TNC implicated in fibrosis persistence [30], and TAPBP which is a subunit of the MHC Class I-TAP complex [31] expressed in stem cells [32,33] (Table 2). Moreover, this shift was corroborated by the reduced expression of proteins encoded by three interferon-stimulated genes (IFI44, IIGP1 and IFIT1) from 7 to 15 dpi (Table 2), as well as the reduction in GO categories related to innate immunity at 15 dpi (Appendix A). The top 5 down-regulated proteins were CBR2 (8.7-fold), TPPP3 (7.1-fold), MB (5.2-fold), PVALB (4.8-fold) and CAVIN2 or SDPR (4.5-fold) (Figure 4B). The continued reduction in expression of CBR2, muscle-associated MB and PVALB concurred with the observation that the lungs were still damaged even at 15 dpi (Figure 1G). Thus, the regenerative process may still be at the early stage in view of the diminished expression of TPPP3 associated with proliferation [34], and SDPR associated with angiogenesis [35]. GO analysis of the down-regulated proteins at 15 dpi also correlated with the persistence of lung injury, where cell adhesion-related functions were still down-regulated (Appendix A).

## 4. Discussion

The lung is a very complex tissue, beginning from the trachea, branching into bronchi that divide into smaller bronchioles, and finally ending as alveoli. In view of its intricate structure, it is likely that there are different populations of stem cells that are separated spatially and temporally, and that respond to specific types of injury to mediate the lung regeneration process [45]. For example, in pneumonia, where the distal alveolar epithelial cells are damaged by infection and/or the concomitant inflammation, it is unlikely for basal cells in the trachea to repopulate the alveolar epithelial cells due to spatial constraints. Indeed, studies identified potential stem cells in the distal area of the lungs that are able to differentiate and give rise to alveolar epithelial cells following various injuries. One population of stem cells constitutes DASCs that express P63 and KRT5 [17,19]. A KRT5 lineage-tracing model was utilized to isolate DASCs which were transplanted into syngeneic mice infected with influenza. The transplanted DASCs could differentiate into alveolar type I and II cells in the recipient mice. Proliferating AT2 cells are considered to be another stem cell population [13]. An SPC lineage-tracing model revealed that AT2 cells proliferated following injury, and eventually differentiated to generate type I cells. This was further substantiated in an influenza pneumonia model [16].

In our sub-lethal influenza pneumonia model, we observed the formation of DASCs (expressing P63 and KRT5) that started migrating out from the bronchioles from 9 dpi, giving rise to obvious DASC pods at 11 dpi, which was also previously reported [19]. However, these DASCs did not co-express alveolar epithelial cell markers at 25 dpi, when the mice had appeared to recover from pneumonia. In fact, the earliest time-point that DASCs were found to have differentiated into AT2 cells was 90 dpi [17,19]. Therefore, it was possible that the DASCs had yet to differentiate into AT2 cells by 25 dpi in our model. However, another study demonstrated that these DASCs did not conclusively differentiate into AT2 cells even up to 200 dpi, but instead evolved into cysts lined with KRT5-positive cells [20]. Taken together with our findings, this suggests that DASCs may not serve as a major source of alveolar epithelial cell renewal in the lungs following influenza-induced pneumonia.

We next addressed whether there are other stem cell populations that can replace the AT2 cells lost during the initial phase of infection, since they were not replenished by DASCs. Interestingly, the population of proliferating AT2 cells increased, even as the animals were recovering from their weight loss. These cells are typically present in low numbers in the alveoli under normal conditions [13], and the sub-lethal infection would be unlikely to damage all AT2 cells. Hence, existing proliferating AT2 cells that are already near or within the damaged area in the lungs may likely be the “first responder” in the lung regeneration process. Indeed, our data suggested that these proliferating AT2 cells that survived the damage induced by infection and inflammation would then begin to divide to generate new AT2 cells. These proliferating AT2 cells eventually stopped dividing once sufficient numbers of AT2 cells were regenerated, thus explaining the reduction in the overall number of cells co-expressing SPC and PCNA proliferation marker, but an increase in cells expressing SPC only.

An interesting question to address is the relationship between proliferating AT2 cells and DASCs. Our data and lineage-tracing evidence [16] suggested that these two cell populations were independent of each other. Although pneumonia may result in severe acute diffuse alveolar damage in the lungs, we found that infected mice were able to recover relatively quickly. Alveolar epithelial cells are crucial in mediating gaseous exchange, with AT2 cells secreting surfactant to reduce surface tension and increase lung compliance, while type I cells facilitate gaseous diffusion. To achieve recovery from pneumonia would require the timely replenishment of both alveolar epithelial cell types in the lungs. Although both DASCs [17,19] and AT2 cells [13] have been shown to give rise to both type I and II cells, our data suggested that the proliferating AT2 cells were the major drivers of alveolar epithelial cell regeneration during the early recovery phase of influenza pneumonia. Based on the spatial distribution of DASCs, it is possible that the role of DASCs was to expand, to line the alveolar spaces to fill the gaps in alveoli arising from the destruction of type I and II cells, in order to mitigate the infiltration of inflammatory cells and fluid infiltrating into the alveolar air spaces. The fact that the DASCs only appeared during severe but not in milder lung injury [20] lends credence to the possibility that the DASCs act as an “emergency repair” mechanism during severe lung injury to counter the inflammation-induced damage. This may also account for the independence of the two cell populations. While it is possible that DASCs may differentiate into type I and II cells during the later phase until complete recovery, we did not observe this phenomenon at least in the immediate repair phase following pneumonia.

It is also noteworthy that these cells involved in lung repair only emerged when the inflammatory process began to shift towards adaptive immune responses. Inflammation is a vital process triggered by infection, and can be divided into three different phases: (a) pro-inflammatory response; (b) switch towards tissue repair; and (c) tissue homeostasis [46]. At 7 dpi when innate immune responses dominated, the DASC pods were not yet formed, and the AT2 cells had not started to proliferate. This is unsurprising in view of the hostile microenvironment expected at the site of inflammation and innate immunity (such as the release of reactive oxygen species and hypoxia), rendering it unfavorable for the cells to begin the repair process. Towards the later stages of pneumonia at around 15 dpi when there was a shift away from the innate immune response, we observed peak AT2 cell proliferation and distinct DASC pods. Inflammation is a key driver of lung tissue regeneration [47,48,49] through the production of growth factors by lung macrophages such as hepatocyte growth factor or HGF [50,51,52,53,54]. Such growth factors promote proliferation and differentiation of the relevant cells [55,56], and may explain the appearance of DASCs and proliferating AT2 cells, during the transition of inflammation towards the tissue repair phase. From this, we can appreciate the importance of balancing of the immune response before and during the recovery phase, and the relationship between inflammatory and regenerative processes in the lungs. Hitherto, it is still unclear how inflammation drives the emergence of these DASCs and proliferating AT2 cells to initiate the tissue repair process. Moreover, the appearance of DASCs is documented in various lung injury models such as bleomycin (to represent acute inflammation) [18] and idiopathic pulmonary fibrosis (chronic inflammation) [57], but not during the normal alveolar turnover process. While inflammation may considerably influence the appearance of DASCs, the degree of lung injury may also determine whether DASCs are required for the repair process [20].

Spatio-temporal quantification of lung cells can enhance our understanding of the dynamic cellular processes of recovery from influenza pneumonia [58]. While we have described the spatial and temporal characteristics of DASCs, and how the proliferating AT2 cell population reacted to alveolar damage induced by influenza pneumonia, there remain unanswered questions concerning alveolar regeneration.

Firstly: Is there a link between the resultant inflammation and the appearance of stem cells in the lungs? One limitation of our study was that the proteomic analysis of the lungs during recovery was conducted on whole lung tissue. Any molecular processes related to regeneration may be masked by the extensive inflammatory processes occurring at the same time. Notwithstanding this, our data indicated that inflammation is a critical process in pulmonary regeneration, consistent with reports describing the vital role of macrophages in alveolar regeneration following infection and pneumonectomy [59,60]. There is increasing evidence that lung resident macrophages contribute significantly to the development of the adult stem cells in the lungs in response to influenza pneumonia.

Secondly: Would the proliferation of AT2 cells be sufficient for lung repair for varying degrees of severity of influenza injury? In our model, we employed the PR8 strain that is specifically mouse-adapted and known to cause severe damage to murine lungs [61]. However, DASCs were not observed using another influenza virus (H3N2 strain X31) which resulted in less severe damage to the lungs [20]. Given that different influenza virus subtypes result in different disease severity [62], the severity of damage may drive differential epithelial regeneration and repair processes, an area worthy of future exploration.

Thirdly: What is the relationship between the proliferating AT2 cells, DASCs and the newly regenerated alveolar epithelial cells during the entire regenerative process? We only described the reported stem cells in the lungs up to 25 dpi, which could be considered to be an early phase of recovery due to the simultaneous presence of lung damage and regeneration. An extended study up to 60 dpi or even longer to study the mid- to even late stage of recovery and tissue repair would elucidate how the different stem cells interact and contribute to alveolar regeneration following influenza pneumonia. Finally, the influence of crucial host factors such as aging and DNA repair processes on lung regeneration also merits consideration [63,64].

## 5. Conclusions

Our data described the spatial and temporal characteristics of DASCs and the characterization of proliferating AT2 cells following influenza virus infection of the lungs. We reported that during the early repair phase up to 25 dpi, DASCs did not give rise to AT2 cells. Instead, the type II cells proliferated to replace those that were lost during infection and inflammation-induced damage. Evidence also suggested that the DASCs may differentiate into some alveolar type I cells directly. The stem cells associated with regeneration did not appear during the period when innate immune responses dominated, instead only appearing when there was a shift towards adaptive immune responses.

## Figures and Tables

**Figure 1 cells-08-00975-f001:**
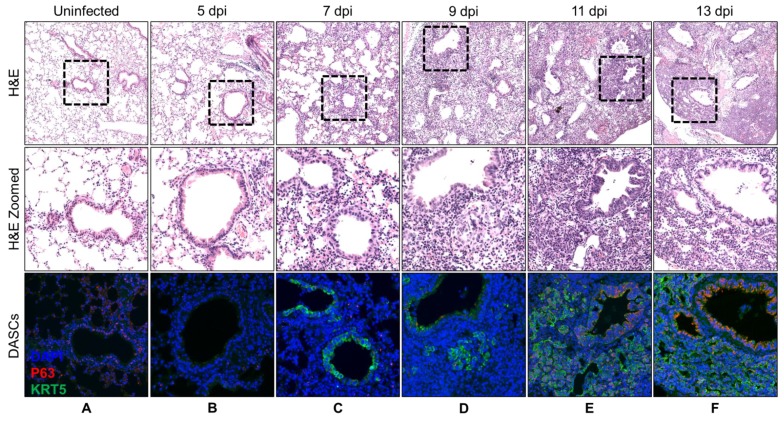
Histopathology and corresponding immunofluorescence staining for P63 and KRT5 within infected mouse lungs over a period until 25 dpi. The top row shows representative H&E images of infected mouse lungs at various time-points following infection. The middle row depicts the zoomed-in images of the top row, while the bottom row portrays the corresponding immunofluorescence staining for P63 and KRT5. No P63-KRT5-positive cells were detected in control uninfected mouse lungs (**A**). DASCs first appeared as small peribronchiolar pods at 9 dpi, before radiating outwards as distinct KRT5-positive pods (**B**–**E**). Formation of lumens was observed by 13 dpi (**F**). DASCs were observed to flatten out and to line the new lumens by 19 dpi (**G**–**I**), and this trend persisted until 25 dpi (**J**–**L**; summarized in Table 1).

**Figure 2 cells-08-00975-f002:**
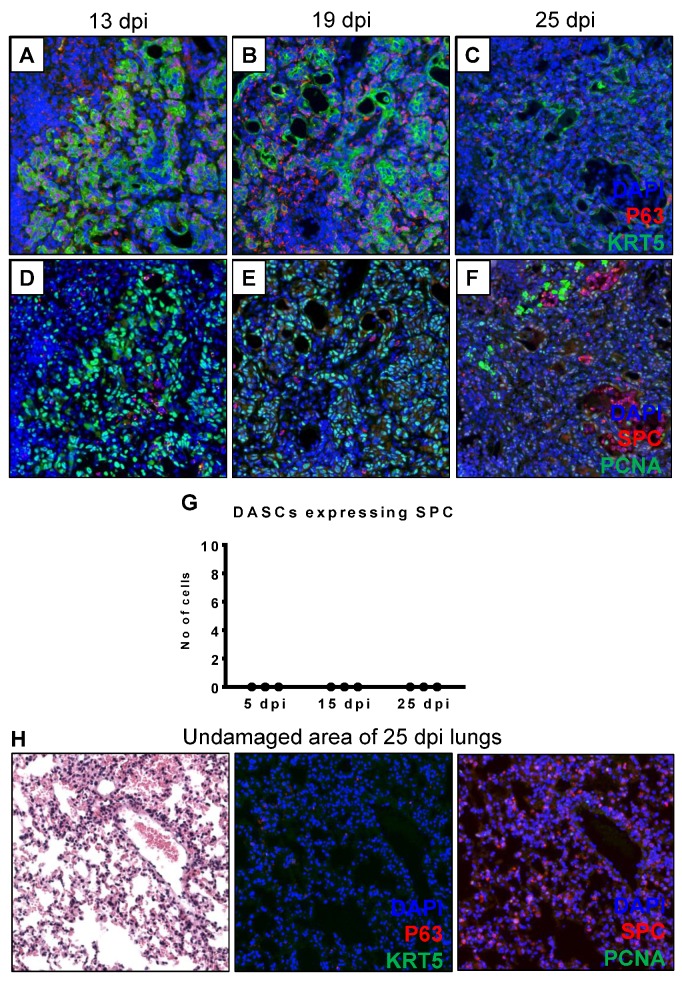
DASCs do not express SPC, the alveolar type II epithelial cell marker. At all three time-points of 13, 19 and 25 dpi, DASCs did not express SPC, suggesting the absence of the differentiation process (**A**–**F**), quantified in (**G**). Some SPC (red) immunofluorescence signal was detected at 25 dpi (**C**), but this was background staining since it appeared in the bronchiolar air-spaces. Decreased proliferation of the DASCs was also observed from 13 to 25 dpi, based on the intensity of PCNA staining (**D**–**F**). To serve as a control, the undamaged area of lungs at 25 dpi expressed SPC, but did not express P63, KRT5 and PCNA (**H**).

**Figure 3 cells-08-00975-f003:**
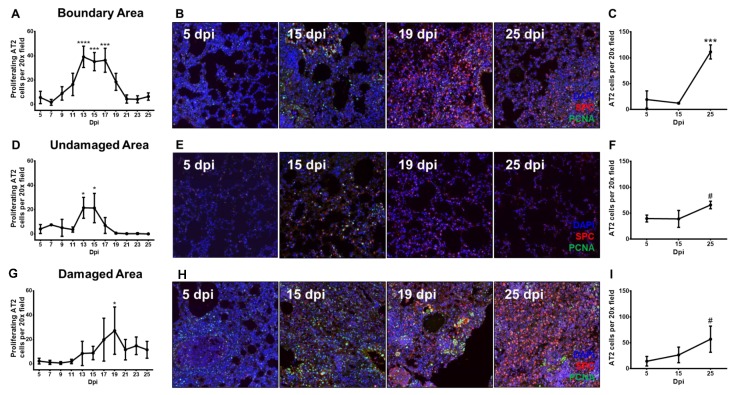
Spatial and temporal quantification of proliferating AT2 cells reveal significant trends during recovery from influenza pneumonia. The number of proliferating for both BA and UA, the peak number of proliferating cells was significantly higher in the BA (**A**,**B**,**D**,**E**). Proliferating AT2 cells in the damaged area (DA) increased from 13 dpi, and peaked at 19 dpi (**G**,**H**). From 21 to 25 dpi, the number of these cells in DA remained higher than BA and UA, whose numbers had diminished to less than 10 cells per field (**A**,**D**,**G**). AT2 cell numbers exhibited an increasing trend from 15 to 25 dpi for BA (**C**), UA (**F**) and DA (**I**), suggesting the AT2 cells were being replenished through AT2 proliferation. * indicates *p* < 0.05, *** indicates *p* < 0.001, **** indicates *p* < 0.0001, # indicates *p* < 0.1. Statistical analysis was conducted by Student’s *t*-test, comparing the cell number of each time-point of each area against that of the respective area at 5 dpi. Error bars were calculated as the standard deviation from three different mice per time-point (5 fields per mouse lung).

**Figure 4 cells-08-00975-f004:**
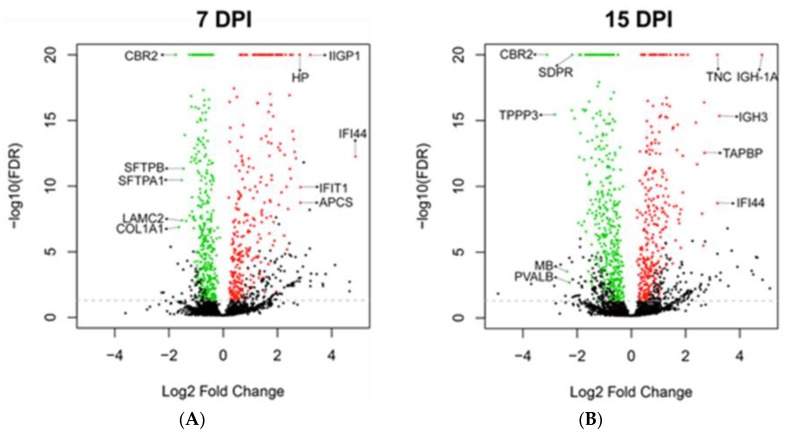
Major up- and down-regulated proteins in whole infected mouse lungs at 7 and 15 dpi. At 7 dpi (**A**), the top 5 out of 271 up-regulated proteins were IFI44, IIGP1, IFIT1, APCS and HP, which are mainly related to innate immunity. The top 5 out of 336 down-regulated proteins were CBR2, COL1A1, SFTPA1, LAMC2, and SFTPB, implying alveolar epithelial damage. At 15 dpi (**B**), the top 5 out of 298 up-regulated proteins were IGH-1A, IGH-3, IFI44, TNC and TAPBP, suggesting the transition to the adaptive immune response. The top 5 out of 393 down-regulated proteins were CBR2, TPPP3, CBR2, MB and PVALB, reflecting persistent damage in the lungs. The functions of the proteins are summarized in Table 2. Proteins coded in black depict non-significant proteins (FDR > 0.01, and number of peptides < 5). The fold change in the expression of each protein is relative to that of control uninfected mice.

**Table 1 cells-08-00975-t001:** Histology and KRT5 expression of infected mouse lungs over time.

Day	Histology Description	KRT5 Expression
5	No recovery.	No KRT5+ cells.
7	No recovery.	KRT5+ cells at bronchioles.
9	Small pods ^1^ seen adjacent to bronchioles.	Small KRT5+ pods near bronchioles.
11	Epithelial regeneration seen as obvious pods as well as single cells radiating out from airway.	Obvious KRT5+ pods.
13	Epithelial regeneration seen as mainly pods with formation of a few lumens.	Obvious KRT5+ pods.
15	Epithelial regeneration seen in all damaged alveoli. Most of the pods were beginning to show lumens.	Obvious KRT5+ pods.
17	Similar to 15 dpi. More pods showing lumens.	KRT5 expression still strong.
19	Lumens began to flatten out and to line alveolar spaces.	KRT5 expression still strong.
21	Progression from 19 dpi.	KRT5 staining intensity reducing.
23	Progression from 21 dpi.	KRT5 staining faint.
25	Progression from 23 dpi.	KRT5 staining even fainter.

^1^ Pods are defined as compact groups of DASCs expressing KRT5 and/or P63 and without a lumen.

**Table 2 cells-08-00975-t002:** Functions of major up- and down-regulated proteins at 7 and 15 dpi.

Day	Regulation	Gene	Function	7 dpi	15 dpi
**7**	Up	IFI44	**Interferon alpha induced protein**. Anti-proliferative activity [36].	+29.4	+9
		IIGP1	**Interferon inducible GTPase 1**. Contributes to resistance against intracellular pathogen [37].	+9.3	+6.4
		IFIT1	**Interferon inducible antiviral RNA-binding protein**. Antiviral activity against influenza virus [38].	+7.2	+2.9
		APCS	**Serum amyloid P-component**. Acute phase protein [39,40].	+7.2	+3.5
		HP	**Haptoglobin**. Acute phase protein [41].	+7.1	N.D.
**7**	**Down**	CBR2	**Carbonyl reductase**. Involved in pulmonary metabolism and highly expressed in alveolar type II cells [42].	−3.4	−8.7
		COL1A1	**Collagen alpha-1(I) chain**, part of type I collagen. Part of lung extracellular matrix [43].	−3.1	−1.9
		SFTPA1	**Surfactant protein A1**. Marker of alveolar type II cells.	−2.9	−2.9
		LAMC2	**Laminin subunit gamma-2**. Subunit of laminin, essential for adherence of epithelial cells to basal membrane [44].	−2.9	N.D.
		SFTPB	**Surfactant protein B.** Marker of alveolar type II cells.	−2.8	−1.6
**15**	Up	IGH-1A	**Immunoglobulin gamma 2A chain C region.**	+4.7	+28.2
		IGH-3	**Immunoglobulin gamma 2B chain C region.**	+3.5	+9.5
		IFI44	**Interferon alpha induced protein**. Anti-proliferative activity [36].	+29.4	+9.0
		TNC	**Tenascin C**. Implicated in fibrosis persistence [30].	+2.2	+9.0
		TAPBP	**Tapasin**. Essential for MHC Class I-TAP complex synthesis [31]. Lowly expressed in stem cells [32,33].	+6.4	+6.5
**15**	Down	CBR2	**Carbonyl reductase**. Involved in pulmonary metabolism and highly expressed in alveolar type II cells [42].	−3.4	−8.7
		TPPP3	**Tubulin polymerization promoting protein**. Associated with proliferation [34].	−2.7	−7.1
		MB	**Myoglobin** and **parvalbumin**. Associated with muscles.	N.D.	−5.2
		PVALB		N.D.	−4.8
		SDPR	**Caveolae-associated protein 2**. Associated with angiogenesis by regulating eNOS activity [35].	−2.4	−4.5

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
