# Peer review of "Insights into Early Recovery from Influenza Pneumonia by Spatial and Temporal Quantification of Putative Lung Regenerating Cells and by Lung Proteomics"

_cells, 2019, doi:10.3390/cells8090975_

Round 1

Reviewer 1 Report

In the present manuscript, the authors investigate the location of, and role in repair for, distal airway stem cells (DASC) and alveolar type II cells (AT2) during influenza-induced lung injury. The data presented are consistent with previous studies demonstrating the appearance and expansion of both cell types following infection. The data are novel and significant in that they establish AT2 cells, rather than DASCs, as the main population responsible for the renewal of alveolar epithelial cells following the resolution of flu infection. Additionally, the author's proteomic data show that AT2 renewal is favored after the innate response has given way to the adaptive response.

My comments and questions:

With some figures the error bars are quite large (e.g., Figures S1, S2, S5, and S6). Do the authors have an explanation for this. Is there confidence that all mice were successfully infected and/or received a similar inoculum?  

As the authors link repair to the transition to adaptive immunity, can they comment on what would occur following a heterosubtypic infection (e.g., X31 H2N2 virus)? Would they expect sooner AT2 proliferative responses?

Author Response

Reviewer 1

In the present manuscript, the authors investigate the location of, and role in repair for, distal airway stem cells (DASC) and alveolar type II cells (AT2) during influenza-induced lung injury. The data presented are consistent with previous studies demonstrating the appearance and expansion of both cell types following infection. The data are novel and significant in that they establish AT2 cells, rather than DASCs, as the main population responsible for the renewal of alveolar epithelial cells following the resolution of flu infection. Additionally, the author's proteomic data show that AT2 renewal is favored after the innate response has given way to the adaptive response.

Response: We thank the reviewer for the helpful comments and suggestions to improve the manuscript.

My comments and questions:

With some figures the error bars are quite large (e.g., Figures S1, S2, S5, and S6). Do the authors have an explanation for this. Is there confidence that all mice were successfully infected and/or received a similar inoculum?  

Response: In the Supplementary Figures S1, S2, S5 and S6, only some time-points have relatively large error bars which may be attributed to experimental variation from animal to animal. In the in vivo experiments, we ensured that the mice were all infected intra-tracheally from the same stock of influenza virus PR8 (20 PFU in 25 μL volume) by the same operator on the same day. However, the severity of infection may differ somewhat from mouse to mouse, which is beyond experimental control. To verify that the mice were successfully infected, we harvested RNA from representative lungs at 5 dpi (which coincides with the peak time-point of greatest viral load), and found that influenza viral NS1 gene could be detected by qPCR. These points have been incorporated into the relevant Methods section.ar.

As the authors link repair to the transition to adaptive immunity, can they comment on what would occur following a heterosubtypic infection (e.g., X31 H2N2 virus)? Would they expect sooner AT2 proliferative responses?

Response: Based on our data, AT2 proliferative responses occur following the transition to adaptive immunity. It has been shown that different influenza virus subtypes result in different disease severity (Rutigliano et al., 2014), e.g. the H3N2 X31 strain results in less severe alveolar damage compared to PR8 strain (Kanegai et al., 2016). In the Discussion section, we have incorporated the following sentence: “Given that different influenza virus subtypes result in different disease severity [62], the severity of damage may drive differential epithelial regeneration and repair processes, an area worthy of future exploration.” We speculate that during an infection that results in milder damage to the lungs, one would expect a more rapid AT2 proliferative response.

Reviewer 2 Report

Comments

The manuscript "Insights into Early Recovery from InfluenzaPneumonia by Spatial and Temporal Quantificationof Putative Lung Regenerating Cells and by LungProteomics" shows the lung recovery within 25 days of a severe influenza infection by alveolar type II cells rather than distal airway stem cells . 

This descriptive analysis of the recovery phenomenon, although carried out on the murine model that does not reach consensus in the field of influenza, provides interesting data.

Proteomic analysis summarizes the steps of the immune response (innate and then adaptive response) and the regeneration process, but it does not answer the question of the molecular processes that induce the repopulation of the alveoli. The discussion clearly reflects the limitations of the techniques used.

Moreover, as point out in discussion, different strains of influenza virus do not induce the same injuries, and hence virulence factors (NS1 for example) of the influenza strain used in the study probably play a predominant role in the duration and in the molecular pathways involved for the regeneration process.

Minor recommendations:

1 – Add a scale bar on histologic and immuno-stained pictures

2 – line 127: double "the" KRT5, remove one

3 – line 136: remove the space in "each lung"

4 - line 145: sequencing instead of "sequecing"

5 - line 148: space between "34" and "were"

Author Response

Reviewer 2

Comments

The manuscript "Insights into Early Recovery from Influenza Pneumonia by Spatial and Temporal Quantification of Putative Lung Regenerating Cells and by Lung Proteomics" shows the lung recovery within 25 days of a severe influenza infection by alveolar type II cells rather than distal airway stem cells .

This descriptive analysis of the recovery phenomenon, although carried out on the murine model that does not reach consensus in the field of influenza, provides interesting data.

Proteomic analysis summarizes the steps of the immune response (innate and then adaptive response) and the regeneration process, but it does not answer the question of the molecular processes that induce the repopulation of the alveoli. The discussion clearly reflects the limitations of the techniques used.

Moreover, as point out in discussion, different strains of influenza virus do not induce the same injuries, and hence virulence factors (NS1 for example) of the influenza strain used in the study probably play a predominant role in the duration and in the molecular pathways involved for the regeneration process.

Response: We thank the reviewer for the helpful comments and suggestions to improve the manuscript.

Minor recommendations:

1 – Add a scale bar on histologic and immuno-stained pictures

2 – line 127: double "the" KRT5, remove one

3 – line 136: remove the space in "each lung"

4 - line 145: sequencing instead of "sequecing"

5 - line 148: space between "34" and "were"

Response: We have made the minor amendments as recommended. The immunofluorescence and histology images were obtained using a scanning microscope at 20x magnification. Following this, we zoomed into the relevant areas of interest at higher magnification. Therefore, we are not able to include a scale bar for the images.